# SoftStep: learning instance-wise similarity functions between neural representations

**Aviad Susman**[*]
aviad.susman@mssm.edu

**Baihan Lin**[†]
baihan.lin@mssm.edu

**Mayte Suárez-Fariñas**[*]
mayte.suarezfarinas@mssm.edu

**Joseph Colonel**[†]
joseph.colonel@mssm.edu

## Abstract

Deep learning systems which make use of similarities between samples mostly rely on fixed or global measures, such as cosine distance and kernels, to compare data points or neural representations. To address this inflexibility, we introduce SoftStep, a module for learning instance-wise similarity measures directly from data. SoftStep maps raw similarity scores into context-sensitive values in the closed unit interval, interpolating smoothly between hard rejection and full preservation of neighbors. Unlike existing approaches such as contrastive learning, sparse attention, or hand-designed geotopological (GT) transforms, SoftStep provides a flexible and learnable mechanism that adapts to the local geometry of the representation space. We demonstrate SoftStep's potential in prediction by incorporating it into a neighbor-based predictor, where it improves performance on multiple datasets. Looking forward, we argue that SoftStep offers a principled extension to topological Representational Similarity Analysis (tRSA) for neural alignment, enabling models to learn GT-like similarity transformations for enforcing alignment. This positions SoftStep as a general tool for learning similarity functions in neural networks.

## 1 Introduction

Similarity functions are an essential tool across machine learning. From contrastive and triplet losses in metric learning [1, 2], to sparse attention mechanisms in transformers [3–5], to representational similarity analysis (RSA) [6] for studying the similarity and semantics of neural representations, comparisons between instances shape the structure of learned representations. Yet, in most cases, these similarity functions are fixed (e.g., cosine similarity, Pearson correlation) or hand-tuned (e.g., geotopological (GT) transforms in tRSA [7]). This rigidity limits the ability of models to adapt similarity judgments to the local geometry of data.

To address this rigidity, we propose SoftStep, a general method for learning instance-wise similarity functions. SoftStep defines a smooth, parametric family of functions that transform raw similarity scores into adaptive similarity values. By learning these transformations jointly with a downstream

---

[*]Department of Population Health Science and Policy, Icahn School of Medicine at Mount Sinai, New York, NY 10029

[†]Windreich Department of Artificial Intelligence and Human Health, Icahn School of Medicine at Mount Sinai, New York, NY 10029

Preprint.

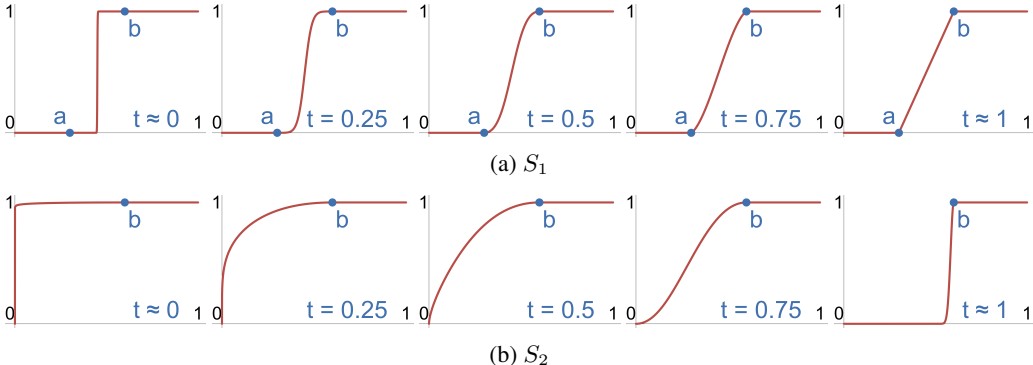

(a) $S_1$

(b) $S_2$

Figure 1: Shape of the $S_1$ and $S_2$ SoftStep functions as the transition parameter $t$ varies from 0 to 1. Both $S_1$ and $S_2$ are families of increasing surjective functions mapping the unit interval to itself.

objective, models can capture nuanced relationships by rejecting task-irrelevant neighbors, preserving close ones, and smoothly interpolating in ambiguous regions.

We first demonstrate SoftStep in a predictive setting, where it improves performance on a neighbor-based regressor. However, the broader implications of SoftStep lie in its generality. In particular, we highlight its natural role in neural alignment [8, 9], where its usage can learn GT-like similarity transformations for tRSA. This affords the possibility of aligning model representations with biological data in a flexible, generalizable, and data-driven manner.

## 2    Methods: SoftStep Similarity Functions

SoftStep is a framework for learning differentiable, parameterized, step-like functions that transform raw similarity scores into context-sensitive similarity values in the range $[0, 1]$. The central innovation is that the parameters controlling the transform, denoted $(a, b, t)$, are not fixed hyperparameters, but rather are learned functions of the anchor point to which similarity is being measured.

In practice, SoftStep can be instantiated using one of two monotonic function families, denoted $S_1$ and $S_2$. Both families define smooth, surjective mappings from $[0, 1] \rightarrow [0, 1]$, but they differ in how they handle thresholds and saturation. The choice between $S_1$ and $S_2$ allows SoftStep to flexibly model either (i) soft thresholding behavior, or (ii) saturating similarity transformations.

Let $z \in \mathbb{R}^d$ denote the embedding of an anchor point, and let $z'$ denote the embedding of a neighbor. We first compute a raw similarity score $s = \text{sim}(z, z')$, normalized to the unit interval $s \in [0, 1]$.

The SoftStep parameters are then defined as $(a, b, t) = g(z)$, where $g : \mathbb{R}^d \rightarrow (0, 1)^3$ is a small learnable network (e.g., a linear projection with sigmoid activation, or a shallow MLP). Thus, the thresholds $(a, b)$ and the smoothness parameter $t$ depend on the location of the anchor point in representation space.

This conditioning enables SoftStep to adaptively warp similarities relative to each anchor in response to factors such as local data density at the anchor point.

### 2.1    SoftStep function families

Given similarity $s$ and parameters $(a, b, t) = g(z)$, SoftStep can be parameterized using one of two families of monotonic transforms:

$$S_1(s; a, b, t) = \begin{cases} 0, & 0 \le x \le a \\ \dfrac{(x-a)^{\frac{1}{t}}}{(x-a)^{\frac{1}{t}} + (b-x)^{\frac{1}{t}}}, & a < x < b \\ 1, & b \le x \le 1 \end{cases} \quad S_2(s; b, t) = \begin{cases} \dfrac{x}{b}^{(b-x)^{\frac{t}{1-t}}}, & 0 \le x < b \\ 1, & b \le x \le 1 \end{cases} \quad (1)$$

| Method | MSE |
|---|---|
| Linear regression head | $5.12 \pm 0.608$ |
| Neighbor-based (w/o SoftStep) | $5.80 \pm 0.444$ |
| Neighbor-based + SoftStep (S1) | $4.06 \pm 0.126$ |
| Neighbor-based + SoftStep (S2) | $4.13 \pm 0.235$ |

Table 1: Mean squared error (MSE) on the RSNA pediatric bone age dataset. Results are reported as mean $\pm$ standard deviation. SoftStep variants achieve the lowest errors. Results are scaled by $10^3$ for readability.

Both families are monotonic surjective mappings from $[0, 1] \rightarrow [0, 1]$, preserving the order of similarities while flexibly re-scaling them.

## 3   Results: SoftStep for Prediction

As a preliminary evaluation of SoftStep as a general similarity mechanism, we first applied it in the context of supervised regression. Specifically, we incorporated SoftStep into a neighbor-based regression head built on top of supervised fine-tuning of pretrained feature extractors [10]. This predictor computes outputs by weighting labels of neighboring instances according to similarities, such as in neighborhood component analysis [11], transformed by SoftStep.

We experimented on the RSNA pediatric bone age dataset [12], which contains 14,000 radiographs labeled by biological age in months. We compared linear layer regression (baseline), neighbor-based regression without SoftStep, and neighbor-based regression with SoftStep $S_1$ and SoftStep $S_2$.

The results (Table 1) show that adding SoftStep reduces mean squared error (MSE) relative to the same architecture without SoftStep, as well as the linear regressor, yielding the best-performing configuration on this dataset. The improvement demonstrates that SoftStep successfully rejects uninformative neighbors and sharpens local similarity structure in the embedding space. Across a broader set of datasets, predictors incorporating SoftStep achieved competitive or superior results, further suggesting that learned similarity transformations generalize across domains (Supplementary Materials).

## 4   SoftStep for Neural Alignment

While regression provides an accessible testbed, the broader motivation for SoftStep lies in neural alignment. Aligning neural network representations to those of biological systems has often relied on representational similarity analysis (RSA) [6], which compares representational similarity matrices (RSMs) using fixed similarity measures such as Pearson correlation.

### 4.1   From RSA to tRSA

Recent work on topological RSA (tRSA) [7] introduced geotopological (GT) transforms, monotonic functions that warp similarity values to better align representational geometries. GT transforms preserve the order of similarities while flexibly re-scaling them, thereby mitigating overfitting to system-specific idiosyncrasies. Experiments show that tRSA is an improvement over traditional RSA for the task of region identification across neural systems. We are motivated by this finding to leverage tRSA to enforce neural alignment.

### 4.2   SoftStep as a Learned GT Transform

SoftStep is a natural generalization of GT transformations, both conceptually and mechanistically. The $S_1$ family of SoftStep functions recovers the GT transform family in the limit as $t \rightarrow 1$, and, more broadly, defines a continuum of smooth similarity transformations. Unlike the original tRSA formulation, which relies on a single global, manually chosen, and non-differentiable transform, SoftStep allows these transformations to be learned end-to-end and adapted instance-wise. This offers

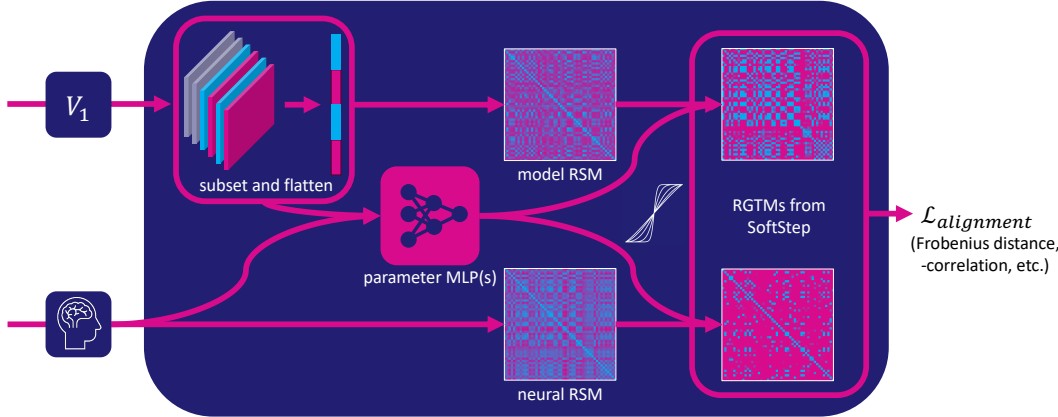

Figure 2: Schematic of the tRSA pipeline with SoftStep modules transforming RSMs from model representations (such as a convolutional block identified with the V1 subregion of the primate VVS) and corresponding neural representations into RGTMs prior to alignment.

two main advantages for neural alignment: (i) differentiability enables SoftStep parameters to be optimized jointly with the model, and (ii) instance-wise flexibility allows different stimuli to induce distinct similarity warps, respecting the fact that how a neural system represents similarity to one stimulus may differ from how it represents similarity to another.

## 4.3  Research Agenda

We propose to integrate SoftStep into tRSA-based alignment between neural network and brain representations. Concretely, given representational similarity matrices $R_{\text{model}}$ and $R_{\text{brain}}$, we will learn and apply SoftStep transforms to both model-side and brain-side similarities before alignment. This allows the alignment process itself to learn which similarities should be sharpened, softened, or discarded. This approach combines the interpretability and neuroscientific grounding of RSA with the flexibility of learned similarity functions, positioning SoftStep as a bridge between predictive deep learning and theoretical neural alignment.

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
