## A   Appendix: SoftStep Regression algorithm

The SoftStep regression algorithm is built on top of neighborhood component analysis [Goldberger et al., 2004]. Given a similarity measure $sim$ an embedded sample $z^*$ and embedded neighbors $Z_N$ we predict

$$\hat{y} = \text{SoftMax}(sim(z^*, Z_N) + \ln(SoftStepPred(z^*, Z_N)))$$

where is a vector of similarities between $z^*$ and the members of $Z_N$ and $SoftStepPred$ is the module described in Algorithm 1.

---

**Algorithm 1** SoftStep for prediction module

---

1: **procedure** SoftStep($Z$, $Z_N$, SoftStep_fn)
    **Initialization (run once at module construction):**
2:    params $\leftarrow$ MLP or Linear layer with sigmoid activation
3:    **Store** SoftStep_fn
    **Forward Pass:**
4:    $(a_0, b_0, t) \leftarrow$ params($Z$)
5:    $sim \leftarrow$ Sim($Z$, $Z_N$)                     ▷ Matrix of similarities
6:    $sim\_norm \leftarrow$ Norm($sim$)             ▷ Normalize to [0, 1]
7:    **if** SoftStep_fn requires $a$ **then**
8:        **if** $Z_N == Z$ (training) **then**
9:            $top\_sim \leftarrow$ row-wise max of $sim\_norm$ excluding diagonal
10:      **else**
11:          $top\_sim \leftarrow$ row-wise max of $sim\_norm$
12:      **end if**
13:      $a \leftarrow \min(a_0, top\_sim) - \epsilon$          ▷ $\epsilon > 0$ small
14:      $b \leftarrow a + b_0 \cdot (1 - a)$
15:    **else**
16:      $a \leftarrow a_0$
17:      $b \leftarrow b_0$
18:    **end if**
19:    $shift \leftarrow$ SoftStep_fn($sim\_norm$, $a$, $b$, $t$)
20:    **return** $sim + \log(shift)$
21: **end procedure**

---

## B   SoftStep prediction experiments

See Table 2 for results.

Exact model versions and pretrained weights are specified in the included GitHub repository. We ensured that at least two distinct feature extractors were chosen per unstructured modality (text, audio, and image) to demonstrate the generalizability of our proposed algorithm. The folowing is a list of datasets and feature extractors used to conduct our experiments:

**RSNA Bone Age Prediction**    The Radiological Society of North America (RSNA) Pediatric Bone Age Machine Learning Challenge collected pediatric hand radiographs labeled with the age of the subject in months Halabi et al. [2019]. We collected 14,036 images from this dataset. Images were resized to 224x224 pixels, normalized with mean and standard deviation of 0.5 across the single gray-scale channel and input to ResNet-18 pretrained on ImageNet He et al. [2016], He and Jiang [2021]. [1]

**ADReSSo**    The Alzheimer's Dementia Recognition through Spontaneous Speech only (ADReSSo) diagnosis dataset has 237 audio recordings of participants undergoing the Cookie Thief cognitive assessment labeled with their score on the Mini Mental State Exam Luz et al. [2021]. Transcripts of these recordings were tokenized and input to DistilBERT-base-uncased Sanh [2019], Zolnoori et al. [2023].[2]

---

[1] https://pytorch.org/hub/pytorch_vision_resnet/
[2] https://huggingface.co/docs/transformers/en/model_doc/distilbert

| Dataset | Linear | SoftStep |
|---|---|---|
| RSNA Halabi et al. [2019] | $5.12 \pm 0.608$ | $4.13 \pm 0.235$ |
| MedSegBench Kuş and Aydin [2024] | $6.69 \pm 1.29$ | $4.02 \pm 0.608$ |
| ADReSSo Luz et al. [2021] | $96.2 \pm 33.3$ | $28.0 \pm 4.13$ |
| CoughVid Orlandic et al. [2021] | $39.7 \pm 27.0$ | $21.2 \pm 0.195$ |
| NoseMic Butkow et al. [2024] | $7.96 \pm 0.947$ | $6.01 \pm 0.472$ |
| Udacity Du et al. [2019] | $0.435 \pm 0.170$ | $0.358 \pm 0.104$ |
| Pitchfork Pinter et al. [2020] | $69.6 \pm 167.0$ | $10.2 \pm 1.01$ |
| Houses Ahmed and Moustafa [2016] | $303 \pm 81.3$ | $13.2 \pm 1.77$ |
| Books (see below) | $8.33 \pm 0.680$ | $7.48 \pm 0.588$ |
| Austin (see below) | $2.29 \pm 0.216$ | $2.05 \pm 0.231$ |

Table 2: Average mean squared error (MSE) $\pm$ standard deviation across ten random splits of each dataset. The best mean results for each dataset are shown in bold. All values are scaled by $10^3$ for readability. A complete description of each dataset, including preprocessing pipelines and feature extractors, is provided.

**MedSegBench**    The MedSegBench BriFiSegMSBench dataset is comprised of single-channel microscopy images and corresponding segmentation masks Kuş and Aydin [2024]. We estimated the size of segmentation mask areas using EfficientNet trained on ImageNet. Rizk et al. [2014], Tan and Le [2019].[3]

**CoughVid**    The CoughVid dataset provides over 25,000 crowdsourced cough recordings, with 6,250 recordings labeled with participant age in years Orlandic et al. [2021]. One second of cough audio was input to HuBERT pretrained on LibriSpeech and mean-pooled across time Hsu et al. [2021], Feng et al. [2024].[4]

**NoseMic**    The NoseMic dataset collected 1,297 30-second audio recordings of heart rate-induced sounds in the ear canal using an in-ear microphone under several activities Butkow et al. [2024]. Audio clips were denoised,[5] encoded with the Whisper tiny audio encoder Radford et al. [2023], and mean-pooled across time.[6]

**Udacity**    The Udacity self-driving car dataset is comprised of dashcam videos labeled with the angle of the car's steering wheel Du et al. [2019]. Videos were downsampled to 4 frames per second for a total of 6,762 images and individual frames were input to EfficientNet trained on ImageNet Tan and Le [2019].[3]

**Pitchfork**    24,649 reviews from the website Pitchfork were collected Pinter et al. [2020], where albums are scored from 0 to 10 in 0.1 increments. 1,500 randomly selected reviews were tokenized and input to BERT base Warner et al. [2024].[7]

**Houses**    The Houses dataset collects 535 curbside images of houses as well as their log-scaled list price Ahmed and Moustafa [2016]. Images were resized to 256x256 pixels, center cropped to 224x224, ImageNet normalized, and input to ResNet-34 He et al. [2016].[1]

**Books**    The MachineHack Book Price Prediction dataset collated 6237 synopses of books labeled with their log-normalized price.[8] Synopses were tokenized and input to DistilBERT-base-uncased.[2]

**Austin**    The Kaggle Austin Housing Prices dataset collects over 15000 descriptions of homes labeled with their log-scaled list price.[9] Descriptions were tokenized and input to DistilBERT-base-uncased.[2]

---

[3] https://docs.pytorch.org/vision/main/models/efficientnet.html
[4] https://huggingface.co/facebook/hubert-base-ls960
[5] https://pypi.org/project/noisereduce
[6] https://huggingface.co/openai/whisper-tiny
[7] https://huggingface.co/google-bert/bert-base-uncased
[8] https://machinehack.com/hackathons/predict_the_price_of_books/overview
[9] https://www.kaggle.com/datasets/ericpierce/austinhousingprices