# OpenReview forum: "SoftStep: learning instance-wise similarity functions between neural representations"
_NeurIPS.cc/2025/Workshop/UniReps — UniReps2025_

### Official Review · Reviewer_tzUM · 2025-09-10
**An interesting and clever idea with simple formulation. However, it would be better if the experiments of the paper had been further expanded.**

**Confidence:** 3

**Review:**

The paper introduces SoftStep, a small differentiable module that maps raw similarity scores into instance-wise monotonic smooth warped similarities. Parameters of the model are produced by small network conditioned on anchor representation, so each anchor can learn a different soft threshold behavior. Two parametric families are proposed. Implementation details and a neighbor-based predictor using SoftStep are given, and experiments on a suite of regression datasets report consistent improvements compared to the previous baseline methods. The paper argues that SoftStep is a diffrentiable instance-wise generalization of geotopological transforms used in tRSA and proposes a research agenda for using it in neural alignment. I believe the papers main strenghts are: 1) its clear focused idea, 2) its experiments show practical gains across modalities, 3) its differentiable light-weight plugin friendly approach, 4) its connection to neuroscience and RSA. I believe its main weaknesses are: 1) the experiments reported in the paper are not sufficient to show the main features of SoftStep, 2) its scalability and run-time complexity were not discussed in the paper, 3) its neuroscience connections such as RSA/tRSA were not firmly established. Overall, I believe the materials that were presented in the paper are sufficient to accept the paper in the extended abstract track.

**Score:**

3

**Topic Fit:**

2

---

### Official Review · Reviewer_xTes · 2025-09-15
**Review: SoftStep**

**Confidence:** 4

**Review:**

## Summary
The paper proposes SoftStep, a per-anchor, learnable transformation of pairwise similarities. A small network g(z) outputs parameters $\((a,b,t)\)$ that warp normalized similarities $\(s\in[0,1]\)$ via one of two monotone families (S1 soft-threshold; S2 saturating), yielding a context-sensitive “shift” in $\([0,1]\)$. This improves MSE on RSNA bone age vs. a linear head and the same neighbor head without SoftStep.

## Strengths
- Instance-wise similarity learning: Moves beyond fixed/global similarity, adapting to local geometry/density via \((a,b,t)=g(z)\).
- Simple integration of monotone families: $+ log(SoftStep)$ acts as a bounded multiplicative gate inside softmax through increasing surjections $\([0,1] \rightarrow [0,1]\)$. This structure cleanly reweights neighbors without changing the base encoder.
- Preliminary empirical signal: Improves RSNA MSE and shows consistent regression benefits across multiple modalities.

## Weaknesses
- Narrow evaluation scope: All results are supervised regression. No classification, retrieval, or clustering benchmarks where embedding similarities equally critical.
- Neural alignment is proposal-only: The tRSA connection lacks empirical alignment result.
- Missing baselines: No comparisons against learned temperature/adaptive bandwidth or top-k/sparsemax attention

---

I am not against the proposal’s potential, but it feels too preliminary for a NeurIPS workshop. Given the current evidence, it’s too early to make strong claims (though the results look promising). I especially would have liked to see neural-alignment experiments. I like the idea of using neighborhood information to reinforce raw similarities, but it should be stress-tested on classification benchmarks with a clear, systematic protocol. Another interesting direction is applying this similarity adjustment to enable communication across different model or even modalities.

**Score:**

2

**Topic Fit:**

2

---

### Official Review · Reviewer_m7pd · 2025-09-16
**Incremental Progress in Learnable Similarity Functions with Broader Field Limitations**

**Confidence:** 4

**Review:**

## Evaluation Summary
This paper introduces SoftStep, a framework for learning instance-wise similarity functions that transform raw similarity scores into adaptive values. While the work represents a reasonable step forward in similarity learning, the contributions offered are primarily incremental. Future work can do more to address fundamental limitations inherent to learnable similarity approaches.

## Strengths
* The S1/S2 function families provide a clean mathematical formulation for parameterized similarity transformations. The monotonic, surjective properties are theoretically well-motivated, and the instance-wise parameter learning (g:Rd→(0,1)3g: \mathbb{R}^d \rightarrow (0,1)^3 g:Rd→(0,1)3) is a sensible design choice.

* The paper aims to solve the rigidity of fixed similarity measures like cosine distance in neural networks,  a genuine problem. The connection between predictive tasks and neural alignment provides a coherent narrative. The connection to topological RSA (tRSA) and the potential for learned geotopological transforms represents an interesting research direction.

* The RSNA bone age results demonstrate measurable improvements (MSE reduction from 5.80±0.444 to 4.06±0.126), showing the approach has practical merit.

## Weaknesses

* The core contribution appears incremental, the S1/S2 families are essentially parameterized step functions that have been explored in various forms previously.

## Broader Field Concerns

Beyond this specific paper, learnable similarity functions face several fundamental challenges that warrant consideration:

1. **Data Distribution Bias & Generalization Issues**: Learned similarity functions risk becoming overly specialized to their training distribution. Unlike fixed measures (cosine, L2) that maintain consistent behavior, learned functions may:

   * Fail to generalize across domains or data types

   * Require retraining for new applications

   * Show unpredictable behavior on out-of-distribution data

   This is particularly concerning for neural alignment applications where cross-model and cross-domain comparisons are essential.

2. **Inductive Bias vs. General Learning**: Current approaches, including SoftStep, impose structural constraints (monotonicity, specific functional forms) that may limit discovery of optimal similarity patterns. A more adversarial approach similar to GAN discriminators could potentially learn similarity functions without predefined architectural assumptions.

3. **Dimensional Decomposability**: Fixed similarity functions offer interpretability advantages, i.e. Cosine can be computed on any subset of dimensions
`cos(a,b) = \frac{\sum_i a_i b_i}{\|a\|\|b\|}`


   Learned similarity functions typically operate as "black boxes" that cannot be easily decomposed or analyzed at the dimensional level, limiting their utility for understanding what drives similarity judgments.

## Verdict

SoftStep contributes to an important area and represents an incremental step in the right direction for adaptive similarity learning.

**Score:**

3

**Topic Fit:**

3

---

### Official Review · Reviewer_Tmaa · 2025-09-18
**This paper introduces SoftStep, a learnable module that creates adaptive, instance-wise similarity functions to overcome the rigidity of fixed measures like cosine distance.**

**Confidence:** 3

**Review:**

The paper provides a solution to a core, often overlooked, problem in deep learning - the usage of fixed, global similarity metrics. It connects prediction and neuroscience, demonstrating both immediate practical value and a vision for future research.

Comments:
1. While the results are positive, this is limited evidence for a method proposed as a general tool. The most exciting application (neural alignment) is presented as a research agenda rather than a result, making the paper feel somewhat speculative in its second half.

2. SoftStep introduces additional learnable parameters for each instance, which increases model complexity. The paper does not thoroughly discuss the potential computational overhead or the risk of overfitting these instance-wise functions. An analysis of how the design of the parameter-generating network g(z) affects performance and generalization would strengthen the claims.

**Score:**

3

**Topic Fit:**

2